# Pterostilbene Induces Pyroptosis in Breast Cancer Cells through Pyruvate Kinase 2/Caspase-8/Gasdermin C Signaling Pathway

**DOI:** 10.3390/ijms251910509

**Published:** 2024-09-29

**Authors:** Tingting Pan, Li Peng, Jing Dong, Lin Li

**Affiliations:** College of Animal Science and Veterinary Medicine, Shenyang Agricultural University, Shenyang 110866, China; ptt1230@stu.syau.edu.cn (T.P.); pengli0735@163.com (L.P.)

**Keywords:** Pterostilbene, breast cancer, metabolism, pyroptosis, GSDMC, caspase-8, PKM2

## Abstract

The incidence and mortality of breast cancer increase year by year, and it is urgent to find high-efficiency and low-toxicity anti-cancer drugs. Pterostilbene (PTE) is a natural product with antitumor activity, but the specific antitumor mechanism is not very clear. Aerobic glycolysis is the main energy supply for cancer cells. Pyroptosis is an inflammatory, programmed cell death. The aim of this study was to investigate the effect of PTE on glycolysis and pyroptosis in EMT6 and 4T1 cells and the specific mechanism, and to elucidate the role of pyruvate kinase 2 (PKM2), a key enzyme in glycolysis, in the antitumor role of PTE. Our study suggested that PTE induced pyroptosis by inhibiting tumor glycolysis. PKM2 played an important role in both the inhibition of glycolysis and the induction of pyroptosis by PTE.

## 1. Introduction

Breast cancer is one of the most common malignancies in women worldwide [1]. More than 20,000 new cases of breast cancer are diagnosed each year, of which nearly 30% result in metastatic disease, causing more than 260,000 deaths annually [1]. Currently, treatment methods for breast cancer include surgery, chemotherapy, radiotherapy, targeted therapy, and immunotherapy [2]. Among these, chemotherapy is the most important systemic treatment for patients with breast cancer [3]. Although chemotherapy for breast cancer has improved significantly in recent years, recurrence and mortality rates remain high [3] due to the complex and diverse underlying mechanisms as well as the emergence of apoptosis resistance [4]. As such, the search for new anti-cancer targets and induction of death pathways other than apoptosis are key to overcoming this problem.

Pyroptosis is a programmed cell death pathway dependent on Gasdermins (GSDMs) [5]. Pyroptosis has so far been shown to occur via five different pathways [6], including inflammasome-dependent and independent pathways. Inflammasome-dependent pathways include the classical caspase-1 pyroptosis pathway and the non-classical caspase-4/5/11 pyroptosis pathway. Independent pathways include the caspase-3/8-mediated and granzyme-mediated pathways [6]. The GSDMs family of porins contains six human genes: GSDMA-E and PJVK [7]. Mice do not express GSDMB but do express triploid GSDMA1–3 and tetraploid GSDMC1–4 [7]. With the exception of PJVK, members of this family contain an N-terminus with pore-forming activity and a C-terminus with self-inhibitory activity [7]. GSDMC is a key factor in newly discovered pyroptosis targeting methods [8]. However, its underlying mechanisms remain poorly explored. Therefore, exploring the mechanism of action of GSDMC in tumors may provide relevant information regarding antitumor therapy.

In cancer cells, glucose is used to produce lactate in the presence of oxygen through a process termed aerobic glycolysis, also known as metabolic reprogramming [9]. In some cases, reprogrammed metabolic activities can be used to diagnose, monitor, and treat cancer [10]. Pyruvate kinase (PK) is a key rate-limiting enzyme in the glycolytic pathway that promotes the conversion of phosphoenolpyruvate and adenosine diphosphate to pyruvate and adenosine triphosphate (ATP) during glycolysis [11]. There are four subtypes of PK: PKM1, PKM2, PKL, and PKR, of which PKM2 is the primary type expressed and upregulated in tumor cells [12]. The highly active tetramer PKM2 is present in the cytoplasm and acts directly as a pyruvate kinase to promote glycolysis [13]. While low-activity dimer PKM2 exists in the nucleus and functions by regulating transcription factors. The enzymatic activity of PKM2 is regulated via a complex mechanism, allowing cells to adapt to different physiological states.

Pterostilbene (PTE) is a stilbene phenolic compound found in various plants [14], and the chemical structure of this compound is shown in Figure 1. PTE has been widely studied owing to its significant antitumor, antioxidant, anti-inflammatory, neuroprotective, and other biological activities [15]. For example, it has been shown to alleviate diabetic complications by targeting glycolysis to exert antioxidant effects [16], in addition to alleviating the inflammatory response in microglia by targeting pyroptosis and exerting neuroprotective effects [17]. However, whether PTE exerts its antitumor effects by targeting glycolysis and pyroptosis remains unclear. As such, it is necessary to further deepen the research on the antitumor effects of PTE to fully understand its antitumor advantages.

## 2. Results

### 2.1. PTE-Induced Pyroptosis in EMT6 and 4T1 Cells

To evaluate the effect of PTE on EMT6 and 4T1 cells, we treated cells with 0, 20, 40, 60, and 80 µM of PTE for 24, 48, and 72 h and then we applied CCK8 to determine the cell viability. The results indicated that PTE significantly inhibited the viability of the EMT6 and 4T1 cells in a dose- and time-dependent manner (*p* < 0.01, Figure 2A). Previous studies confirmed that PTE induced cell death through a variety of death modes [18,19]. Thus, we co-treated cells with specific death inhibitors, including ferroptosis (Fer-1), autophagy (CQ), necroptosis inhibitors (Nec-1), and pyroptosis inhibitors (Z-VAD-FMK), with PTE. Morphological observations revealed that only Z-VAD-FMK improved PTE-induced reduction in cell density (*p* < 0.01, Figure 2B). The results of CCK8 showed no significant increase in cell viability after treatment with PTE with other inhibitors other than Z-VAD-FMK (*p* < 0.01, Figure 2C). These results suggested that only Z-VAD-FMK of these inhibitors alleviated PTE-induced EMT6 and 4T1 cell death. Therefore, we tested indicators related to pyroptosis. The results showed that EMT6 and 4T1 cells were swollen, ruptured, and large bubbles blew out after PTE treatment (*p* < 0.01, Figure 2D). Lactate dehydrogenase (LDH) release and interleukin (IL)-18 and IL-1β activation were increased in the cell supernatant (*p* < 0.01, Figure 2E,F). Most importantly, the key indicators activation of pyroptosis increased after PTE treatment (*p* < 0.01, Figure 2F). In addition, we tested the toxicity of PTE to HC11. The result showed that PTE did not affect the viability of HC11 (*p* > 0.05, Figure 2G) and indicated that PTE was non-toxic to mouse mammary epithelial cells.

### 2.2. PTE-Induced Cancer Cell Pyroptosis Is Mediated by GSDMC

The above experiments suggested that PTE induced pyroptosis in EMT6 and 4T1 cells, but the specific mechanism was not well understood. We measured the mRNA expression of GSDMA, GSDMC, GSDMD, and GSDME in EMT6 and 4T1 cells. In EMT6 cells, the relative mRNA expression of GSDMC was the highest and PTE had the most obvious regulation on GSDMC. In 4T1 cells, the relative mRNA expressions of GSDMC and GSDMD were similar, but the regulation of GSDMC by PTE was more obvious (*p* < 0.01, Figure 3A). Therefore, we hypothesized that PTE will induce pyroptosis in EMT6 and 4T1 cells via GSDMC. The above experiments had demonstrated that PTE activated GSDMC (Figure 2G). In order to more intuitively observe the effect of PTE on the localization of GSDMC proteins, we tested the intracellular localization of GSDMC (*p* < 0.01, Figure 3B). The results showed that GSDMC was diffusely distributed in the cytoplasm of EMT6 and 4T1 before PTE treatment, while there was obvious membrane localization after PTE treatment. The result indicated the presence of pore formation of the GSDMC-NT terminal domain on the cell membrane after PTE treatment. To investigate the role of GSDMC in PTE-induced pyroptosis, we constructed EMT6 and 4T1 cell systems that stably interfered with GSDMC, and we measured the protein expression (*p* < 0.01, Figure 3C). After silencing GSDMC, we found a significant decrease in the number of pyroptosis cells (*p* < 0.01, Figure 3D). At the same time, we evaluated the cell viability, LDH release, and IL-18 and IL-1β actitation (*p* < 0.01, Figure 3E–G). The results showed that silencing GSDMC alleviated PTE-induced pyroptosis in EMT6 and 4T1 cells and inhibited PTE-induced LDH release and IL-18 and IL-1β activation. To sum up, in EMT6 and 4T1 cells, PTE-induced pyroptosis was via GSDMC.

Next, we explored the upstream mechanism of PTE-induced pyroptosis. A previous article reported that GSDMC was cleaved by caspase-8 [20]. So, our next experiment will verify whether the upstream factor of PTE-activated GSDMC was caspase-8 in EMT6 and 4T1 cells. The above experiments demonstrated that PTE activated caspase-8 (*p* < 0.01, Figure 2G). First, we determined the use concentration and time of Z-VAD-FMK and caspase-8 inhibitor (Z-IETD-FMK) through the literature review and WB test (*p* < 0.01, Figure 3H). We used the caspases inhibitor Z-VAD-FMK to confirm whether PTE induced pyroptosis via the caspases’ pathway. The results showed that Z-VAD-FMK significantly inhibited the activation of GSDMC and caspase-8 induced by PTE (*p* < 0.01, Figure 3I). The results suggested that PTE induced pyroptosis by caspases. Therefore, we used Z-IETD-FMK for a more in-depth study. It also inhibited PTE-induced activation of caspase-8 and GSDMC (*p* < 0.01, Figure 3J,K). These results suggested that PTE-induced pyroptosis was dependent on caspase-8.

### 2.3. PTE Inhibits the Glycolysis of EMT6 and 4T1 Cells

The change in glucose concentration in the medium reflected the size of glucose uptake capacity of cells. The amount of lactate production indicated how much glucose enters aerobic glycolysis. When cells die, the ATP decreased dramatically [9]. By colorimetric kit, we found that treatment glucose consumption was reduced, lactate production was weakened, and ATP content was decreased in EMT6 and 4T1 cells after PTE treatment (*p* < 0.01, Figure 4A). In addition, we detected protein expression of key factors critical to glycolysis. The results showed that PTE down-regulated the protein expression of these factors (*p* < 0.01, Figure 4B). These results suggested that PTE effectively inhibited aerobic glycolysis in EMT6 and 4T1 cells.

### 2.4. PTE Inhibits Glycolysis by Regulating PKM2, a Key Rate-Limiting Enzyme in Glycolysis

The above trials demonstrated that PTE had the most significant regulation on PKM2. So, we hypothesized that PTE regulated tumor glycolysis through PKM2. Next, we looked at how PTE regulated PKM2 in EMT6 and 4T1 cells. PKM1 promoted oxidative phosphorylation, and PKM2 encouraged aerobic glycolysis [21]. Our results showed that PTE down-regulated the expression of PKM2 and upregulate the expression of PKM1 in tumor cells (*p* < 0.01, Figure 5A). Glutaraldehyde cross-linking revealed that PTE down-regulated the expression of PKM2 tetramers and dimers (*p* < 0.01, Figure 5B). PKM2 dimer transfered to the nucleus, promoted the activation of a variety of transcription factors, affected a variety of signaling pathways, and promote the development of tumors [13]. Results of WB showed that PTE treatment down-regulated the expression of PKM2 in the nucleus (*p* < 0.01, Figure 5C). Phosphorylation of PKM2 at different sites exhibited different functions. Y105 phosphorylation regulated glycolytic flux by controlling the flux of active and inactive forms of PKM2 and affecting the multimeric form of PKM2 [22]. We found that PTE down-regulated the expression of phosphorylated PKM2 protein at Y105 (*p* < 0.01, Figure 5D). Finally, we found that PTE down-regulated the activity of pyruvate kinase (*p* < 0.01, Figure 5E). In order to investigate the role of PKM2 in glycolysis, we knocked down the expression of PKM2 by small interfering RNA (*p* < 0.01, Figure 5F) and then detected the content of glucose, lactate, and ATP content by the kit. The results showed that knocking down PKM2 alleviated the PTE-induced reduction in glucose consumption, lactate, and ATP production (*p* < 0.01, Figure 5G). In summary, PTE inhibited the glycolysis by PKM2 in EMT6 and 4T1 cells.

### 2.5. PTE-Activated Caspase-8/GSDMC Cascade through Repressing PKM2

To further explore the role of PKM2 in pyroptosis, we combined PTE with PKM2 inhibitors (PKM2-IN-1) or activators (TEPP-46). First, we determined the concentration and time of PKM2-IN-1 and TEPP-46 (*p* < 0.01, Figure 6A). PKM2-IN-1 was co-treated with PTE, the activation of caspase-8, cleavage of GSDMC increased in EMT6 and 4T1 cells, while TEPP-46 had the opposite effect (*p* < 0.01, Figure 6B). Morphological observations showed that pyroptosis increased after PKM2-IN-1 co-treating with PTE, while the TEPP-46 had the opposite effect (*p* < 0.01, Figure 6C). An evaluation of cell viability, IL-18 and IL-1β activation, and LDH release further proved that PKM2-IN-1 had a synergistic interaction with PTE, while TEPP-46 had the opposite reaction (*p* < 0.01, Figure 6D–F). In addition, we also confirmed the interaction of PKM2 and caspase-8 through co-IP tests (*p* < 0.01, Figure 6G). These results confirmed that PTE induced pyroptosis of the caspase-8/GSDMC pathway via PKM2.

### 2.6. PTE Reduces Tumorigenicity in Mouse Breast Tumor Models

Next, we examined the effect of PTE administration on tumor growth in vivo. During the treatment period, the body weight of the mice was relatively stable, and there were no signifi t differences between the groups (*p* > 0.05, Figure 7A), indicating that PTE had a high safety profile in mice. By weighing and observing mice’s tumor appearance, we found that PTE down-regulated tumor size and volume (*p* < 0.01, Figure 7B–D). Furthermore, activation of pyroptosis-related factors, LDH release, and activation of IL-18 and IL-1β increased after PTE treatment (*p* < 0.01, Figure 7E,F). These results indicated that PTE induced pyroptosis in mouse breast xenografts. Colorimetric kit results showed that PTE down-regulated glucose consumption, lactate production, and ATP content (*p* < 0.01, Figure 7G). WB detection showed that PTE could down-regulate the protein expression of glycolysis genes (*p* < 0.01, Figure 7H). These results demonstrated that PTE inhibited the glycolysis of tumor tissue.

## 3. Discussion

The pathogenesis of breast cancer, which poses a significant threat to women’s health, is complex and diverse [23]. The regulation of metabolism and cell death are two important aspects of breast cancer inhibition [24,25]. However, the specific mechanism underlying is not clear. Our study is the first to suggest that treatment with PTE inhibits the expression of PKM2 to trigger pyroptosis in EMT6 and 4T1 cells via the caspase-8/GSDMC pathway (Figure 8). Further, we showed that PTE inhibits the growth of mouse breast xenografts in vivo.

Several studies have previously investigated the antitumor effects of phenolic compounds. For example, research has shown that the commonly used phenolic compounds resveratrol and quercetin can induce apoptosis in a variety of cancers, such as breast, liver, ovarian, and colon cancers [26,27,28,29]. However, the bioavailability of resveratrol is not as high as that of PTE, while the safety of quercetin is lower than that of PTE as it can affect pregnancy and lactation [30,31]. The safety of PTE has also been confirmed in previous studies, in which mice were fed a diet containing low, medium, and high doses of PTE, resulting in no significant changes in histopathology, hematology, clinical chemistry, or urine balance compared to controls [32]. Our results further showed that low, medium, and high doses of PTE were safe for mice in the short term. The safety of PTE at the cellular level has also been demonstrated in several studies [33,34,35]. Moreover, our study indicated that PTE had no effect on HC11 cell viability. Most cells were resistant to apoptosis. As such, it is important to kill tumor cells through other routes of death. Our study confirmed the anti-breast tumor effect of PTE from two aspects: regulation of tumor metabolism and induction of pyroptosis, indicating the potential of PTE as a key antitumor drug acting on multiple pathways. It is worth mentioning that the double bonds in Pterostilbene are photosensitive, making it highly susceptible to photodegradation, which raises stability concerns. Research is also underway to develop new drugs that address this photodegradation issue [36]. 

Previous studies reported that PTE can inhibit the growth of various tumors by inducing cell death [18,19]. For example, PTE can induce apoptosis in gliocytomas [18], human oral cancer autophagy [19]. To investigate how PTE induces EMT6 and 4T1 cell death, we treated cells with several common death inhibitors in combination with PTE. The results showed that pyroptosis inhibitors alleviated PTE-induced EMT6 and 4T1 cell death. This result was then positively confirmed by morphological observations showing LDH release, activation of IL-18 and IL-1β, and protein expression of pyroptosis-related factors. Most studies have further shown that PTE can induce apoptosis or target key tumor signals to inhibit cell proliferation and invasion in most cases [37,38,39,40]. For example, PTE can inhibit tumor growth by targeting estrogen receptors, fighting drug resistance [37,38], and inducing apoptosis through the p53/cyclin E1 [39], Bcl-2/caspase-3/caspase-9 [40], and caspase-8 signaling pathways [41]. Caspase-8 has long been considered a hallmark of apoptosis; however, several studies have indicated that defining caspase-8 as a marker of apoptosis alone is inaccurate. Indeed, caspase-8 can activate both GSDMD-induced [42] and GSDMC-mediated pyroptosis [20]. GSDMC is a newly discovered antitumor pyroptosis target with confirmed significance in clinical research [8]. By silencing GSDMC, we confirmed that PTE-induced pyroptosis was GSDMC-mediated. Z-VAD-FMK is a broad caspase inhibitor, and we confirmed that PTE-induced pyroptosis is closely related to the caspase family of this molecule. Z-IETD-FMK is a specific inhibitor of caspase-8, and using this molecule, we confirmed that the activator of GSDMC is caspase-8. Cell viability, LDH release, and IL-18 and IL-1β activation were subsequently measured. According to these data, PTE-induced pyroptosis of EMT6 and 4T1 cells was confirmed through the caspase-8/GSDMC pathway. In addition, because caspase-8 plays an important role in both apoptosis and pyroptosis, we speculated that apoptosis may be the primary mode of PTE-induced cell death in cells with very low GSDMC expression.

Abnormal energy metabolism associated with high glucose uptake is a unique feature of cancer [43]. Studies have shown that targeted tumor glycolysis promotes tumor cell death [44,45]. Specifically, enhancing glycolysis in colon cancer can promote tumor angiogenesis [44], while the inhibition of glycolysis in bladder cancer can inhibit tumor proliferation [45]. The mechanisms of action of the drug regulation of glycolysis are also being explored. Resveratrol inhibits the proliferation and induces the apoptosis in ovarian cancer cells [46]. Similar to the antitumor principles of the above compounds, we found that PTE could inhibit glycolysis in EMT6 and 4T1 cells based on the detection of key indicators of glycolysis. PKM2 is a key rate-limiting enzyme of glycolysis as well as an important target for tumor inhibition [47,48,49], and inhibiting the expression of PKM2 can suppress the progression of a variety of cancers, including lung [47], breast [48], and colon [49] cancers. In the present study, we found that PTE was the most pronounced regulator of PKM2 expression in EMT6 and 4T1 cells. Therefore, we suspect that PTE inhibits glycolysis through PKM2. Our data show that the tetramers of PKM2 and PKM2 in the nucleus were down-regulated by PTE. After silencing PKM2, glycolysis was inhibited by PTE. Therefore, we believe that PTE inhibits glycolysis in EMT6 and 4T1 cells via PKM2. It is worth mentioning that hexokinase 2 (HK2) and phosphofructokinase-1 are also rate-limiting enzymes in glycolysis and play important roles in glycolysis as PKM2. However, the roles of these regulators in breast cancer remain unexplored.

The relationship between glycolysis and pyroptosis has become increasingly clear. Pyroptosis-related factors are regulated by glycolysis, such as the disruption of glycolytic flux as inflammasome activator [50]. Our study demonstrated the importance of PKM2 in glycolysis in EMT6 and 4T1 cells. Therefore, we hypothesized that PKM2 may play a role in PTE-induced pyroptosis. TEPP-46 activates PKM2 by promoting the synthesis of tetramers and PTE agonists. PKM2-IN-1 suppresses PKM2 activity by restraining its expression in synergy with PTE. Using these two molecules, we found that PKM2 suppression could induce pyroptosis of the caspase-8/GSDMC pathway in EMT6 and 4T1 cells. In addition, GSDMD and GSDME are important targets of pyroptosis, and the role of PKM2 in these two pathways is worth exploring.

In summary, our results suggest that PTE induces pyroptosis in EMT6 and 4T1 cells by inhibiting glycolysis. This not only provides a high-efficiency and low-toxicity natural drug for clinical antitumor activity, but also provides a new antitumor target and direction.

## 4. Methods

### 4.1. Cell Culture

The mouse breast cancer cell lines (EMT6 and 4T1) and mouse mammary epithelial cells (HC11) were purchased from Wuhan Procell Life Science and Technology Co., Ltd. (Wuhan China). The cells were grown in 1640 medium (Thermo, Waltham, MA, USA) enriched with 10% fetal bovine serum (FBS) (Pricella, Wuhan, China) and antibiotics (penicillin and streptomycin, each used in concentrations of 25 U/mL) (Pricella, Wuhan, China). Cells were cultured at 37 °C with 5% CO_2_ in humidified air.

### 4.2. Test Animals

All animal experiments were approved by the Animal Experimentation Committee of the Shenyang Agricultural University. Ethical approval was granted by the Shenyang Agricultural University Animal Experiments Committee. Purchase of 80 healthy, pathogen-free female BALB/C mice aged 6–8 weeks and weighing 18–22 g from Liaoning Changsheng Biotech Co., Ltd. (Liaoning, China). The mice were kept in a 25 ± 2 °C animal house without specific pathogens, and the light and dark light alternated for 12 h every day. Animals were randomly assigned to experimental groups prior to tumor cell injection. Before tumor cell injections, animals were anesthetized in 1.5–2% isoflurane in 100% oxygen. Mice in the control group normal saline gavaged, which was refreshed 2–3 times per week. Mice in the treatment group PTE gavaged, which was refreshed 2–3 times per week. Tumors were measured in three directions and the volume was calculated as (length × width × height)/2.

### 4.3. Preparation Solutions

The PTE used in the experiment was purchased from Shanghai Yuanye Biotech (Shanghai, China). PKM2-IN-1, TEPP-46, and Z-IETD-FMK were purchased from GLPBIO (Montclair, CA, USA). Z-VAD-FMK, Fer-1, CQ, and Nec-1 were purchased from MedChemExpress (Middlesex, NJ, USA). Dimethyl sulfoxide (DMSO) was purchased from Solarbio (Beijing, China). PTE, Z-VAD-FMK, Fer-1, CQ, Nec-1, PKM2-IN-1 TEPP-46, and Z-IETD-FMK mother liquor were prepared with DMSO as solvent and diluted with 1640 medium to the desired concentration for the experiment. The concentration of DMSO did not exceed 0.1% [51]. Z-VAD-FMK was treated with a concentration and time of 10 µM for 24 h. The treatment concentration and time of Fer-1 was 5 µM for 48 h. The treatment concentration of Nec-1 was 100 µM for 48 h, and the use concentration and time of CQ was 5 µM for 48 h [52,53,54]. PKM2-IN-1 was treated at a concentration and time of 65 µM for 10 h. The treatment concentration and time of TEPP-46 was 35 µM for 6 h. Z-IETD-FMK was used at a concentration and time of 45 µM for 10 h.

### 4.4. Cell Counting Kit-8 (CCK8) Assays

Cells were seeded in a 96-well plate. Once cells reached 90% confluence, the growth media were removed, the cells were washed twice in phosphate-buffered saline (PBS, pH 8.0), and media-enriched different drugs were replaced. The specific test groups were as follows: (1) 0, 40, 60, 80, and 160 µM PTE for 24 h. (2) First, 10 µM Z-VAD-FMK for 24 h, 5 µM Fer-1 for 48 h, 100 µM Nec-1 for 48 h, 5 µM CQ for 48 h, and then treated with 40 µM PTE for 24 h. (3) First, silent the GSDMC and then treated with 40 µM PTE for 24 h. (4) First, 65 µM PKM2-IN-1 for 10 h, 35 µM TEPP-46 for 6 h, and then treated with 40 µM PTE for 24 h. Subsequently, CCK8 solution was added to each well for 2 h at 37 °C in the dark. Optical density (OD) was measured using a microplate reader at a wavelength of 540 nm.

### 4.5. Morphological Observations

Cells were seeded in a 6-well plate. Once cells reached 90% confluence, the growth media were removed, the cells were washed twice in PBS, and media-enriched different drugs were replaced. After 24 h, they were taken out, washed once with PBS, and photographed under a microscope. The specific test group was the same as the CCK8 assays.

### 4.6. Real-Time Fluorescence Quantitative PCR (qPCR)

Cells were treated with 1640 medium or 40 µM PTE for 24 h. Total RNA was isolated using trizol reagent (Invitrogen/Thermo Fisher Scientific, Waltham, MA, USA) after 24 h. The extracted RNA (2 µg) was treated with DNase I (Thermo Fisher Scientific, Kalamazoo, MI, USA) and reverse transcribed by using the iScript™ cDNA Synthesis Kit (K1632; Fermentas, Burlington, ON, Canada). qPCR was performed using SYBR Premix Ex Taq (#RR420A; Takara, Otsu, Japan) on a 7500 Fast Real-Time PCR System (Applied Biosystems, Foster City, CA, USA).

### 4.7. Western Blot (WB)

Cells treated with different drugs were collected, and the total protein was extracted using a strong RIPA buffer (Yamei; Shanghai, China) supplemented with PMSF (Yamei; Shanghai, China) and a phosphatase inhibitor cocktail (Yamei; Shanghai, China). The resulting supernatants were then quantified with a BCA kit (Yamei; Shanghai, China). Equal amounts of each protein extract were then separated by SDS-PAGE and transferred onto PVDF membranes (Immobilon; Shanghai, China). After blocking, the membranes were incubated overnight at 4 °C with specific primary antibodies. The following day, the membranes were incubated with appropriate secondary antibodies at room temperature for 1 h. Finally, the target protein bands were visualized using an IBright (Thermo, Waltham, MA, USA) after exposure to the ECL substrate (Biosharp; Changsha, China). The specific test groups were as follows: (1) 1640 for 24 h; (2) 40 µM PTE for 24 h; and (3) first, 10 µM Z-VAD-FMK for 24 h, Z-IETD-FMK 45 µM for 10 h, 65 µM PKM2-IN-1 for 10 h, 35 µM TEPP-46 for 6 h, and then 40 µMPTE for 24 h.

### 4.8. Immunofluorescence

Cells were seeded on glass slides in 6-well plates. Once cells reached 90% confluence, the growth media were removed, the cells were washed twice in PBS, and media 1640 medium or 40 µM PTE were replaced. After 24 h, cells were washed twice with PBS and fixed with 4% paraformaldehyde for about 20 min at room temperature (Biosharp, Shanghai, China). Then, cells were washed in PBS for 15 min. The cells were incubated on ice in 0.5% Triton X-100 (Beyotime, Shanghai, China) for 15 min. The cells were exposed to BSA (Solarbio, Beijing, China) for 20 min. Samples were incubated with primary antibodies (Abclonal, Wuhan, China, 1:100) overnight at 4 °C in a buffer solution. Cells were washed in PBS, followed by incubation with Alexa Fluor™ 568 goat anti-rabbit (Abclonal, Wuhan, China, 1:250) at room temperature (RT) for 1 h and counterstained with 4′,6-diamidino-2-phenylindole (DAPI) (Solarbio, Beijing, China) for 15 min. They were washed in PBS for 15 min, and then the results were observed under a fluorescence microscope.

### 4.9. DSS Cross-Linking Test

For detection of dimeric/tetrameric PKM2 expression, cells treated with 1640 medium or 40 µM PTE were collected, washed twice with PBS, and incubated in 2 mM disuccinimidyl suberate (DSS) (Sangon Biotech, Shanghai, China), which dissolved in PBS (pH 8.0) for 30 min at 37 °C. Cells were then washed with PBS, lysed in RIPA buffer, mixed with SDS loading buffer, and heated at 100 °C for 5 min. The expression of PKM2 dimers and tetramers was detected by WB.

### 4.10. Transfection

EMT6 and 4T1 cells were seeded in a 6-well plate for 50% density the day before transfection. Transfection could be performed when the cells grow to a density of 70%. siRNA was mixed with Lipofectamine 3000 (Invitrogen, Carlsbad, CA, USA) based on the manufacture’s guide, incubated for 10 min at room temperature, and then added into the 6-well plate. Cells were collected 48 h after transfection. Transfection efficiency was detected by WB.

### 4.11. Nucleoplasmic Separation

Nuclear and cytoplasmic fractions of EMT6 and 4T1 cells were separated with a kit (Beyotime, Shanghai, China). The analysis of protein was conducted by WB. The markers for the nucleus and cytoplasm were, respectively, Lamin B and GAPDH.

### 4.12. Pyruvate Kinase Activity

Cells were seeded in a 6-well plate. Once cells reached 90% confluence, the growth media were removed, the cells were washed twice in PBS, and media-enriched different drugs were replaced. The media were collected after 24 h incubation for glucose content assays (Elabscience, Wuhan, China). The supernatant for glucose content assays (Elabscience, Wuhan, China). The specific test groups were as follows: (1) 1640 for 24 h (2) 40 µM PTE for 24 h; and (3) first, 65 µM PKM2-IN-1 for 10 h, 35 µM TEPP-46 for 6 h, and then treated with 40 µM PTE for 24 h.

### 4.13. Glucose, Lactate, and Adenosine Triphosphate (ATP) Detection

Cells were seeded in a 6-well plate. Once cells reached 90% confluence, the growth media were removed, the cells were washed twice in PBS, and media-enriched different drugs were replaced. The media were collected after 24 h incubation. Tumor tissue was homogenate and centrifuged to take the supernatant. We qualified glucose utilization, lactate production, and intracellular ATP production by using glucose assay kit (Beyotime, Shanghai, China), lactate assay kit (Elabscience, Wuhan, China), and ATP assay kit (Elabscience, Wuhan, China), respectively, as per the manufacturer’s instructions.

### 4.14. Co-IP

EMT6 and 4T1 cells were lysed in a lysis buffer (50 mM Tris–HCl, pH 7.4, 0.1% SDS, 150 mM NaCl, 1% Triton X-100, 1 mM NaF, 1 mM EDTA, 1 mM Na_3_VO_4_, and 1× protease inhibitor cocktail) and then immunoprecipitated with PKM2 and caspase-8 antibody. Thereafter, Protein A/G sepharose beads (Biolinkedin, Shanghai, China) were applied. The final precipitated proteins were subjected to immunoblotting with PKM2 and caspase-8 antibody

### 4.15. Statistical Analyses

Statistical analyses were conducted using IBM SPSS Statistics (v. 22) and GraphPad Prism (v.8) software. The data were the result of three independent experiments. The descriptive statistics function was used to generate the mean and SD, and the results were expressed in the error bars. A one-tailed t-test was used to calculate significant differences in *p*-values for the tested samples compared with the control. *p* < 0.05 was considered to indicate statistical significance. Data were mean ± SD, * meant *p* < 0.05, ** meant *p* < 0.01. ^ meant *p* < 0.05, and ^^ meant *p* < 0.01.

## 5. Conclusions

The main findings of the paper are as follows: PTE induced pyroptosis in EMT6 and 4T1 cells and the specific mechanism was the caspase-8/GSDMC pathway. PTE inhibited glycolysis in EMT6 and 4T1 cells via PKM2. In EMT6 and 4T1 cells, PKM2 played an important role in PTE-induced pyroptosis of caspase-8/GSDMC. In our study provided a novel paradigm of antitumor activity of PTE in breast cancer. Furthermore, this study highlighted a previously unrecognized function of PKM2 in suppressing pyroptosis in cancers.

## Figures and Tables

**Figure 1 ijms-25-10509-f001:**
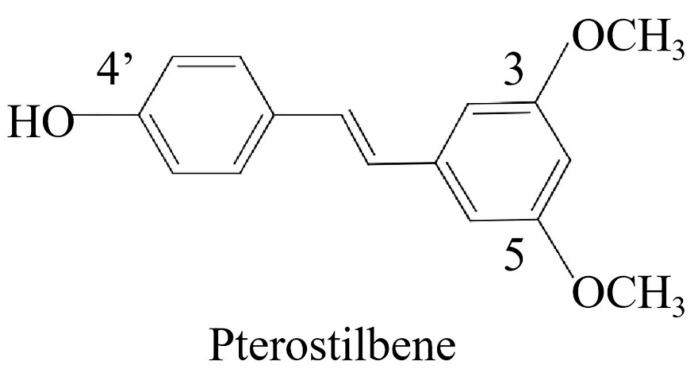
Chemical structure of PTE.

**Figure 2 ijms-25-10509-f002:**
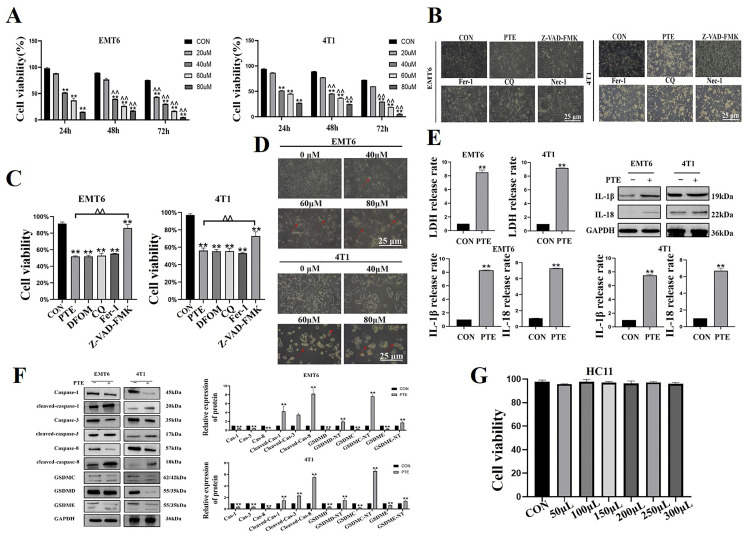
PTE induced pyroptosis in EMT6 and 4T1 cells. (**A**) Effect of PTE on EMT6 and 4T1 cell viability. All of the following experiments treated cells with 40 µM PTE. (**B**) Effects of CQ, Fer-1, Nec-1, and Z-VAD-FMK combined with PTE on morphology of EMT6 and 4T1 cells under a 200-fold field of view. (**C**) Effects of CQ, Fer-1, Nec-1, and Z-VAD-FMK combined with PTE on the viability of EMT6 and 4T1 cells. (**D**) Morphological effects of PTE on EMT6 and 4T1 cells under a 200-fold field of view. The red arrows pointed to cells that showed pyroptosis. (**E**) LDH determination and IL-18 and IL-1β activation. (**F**) Effect of PTE on the expression of pyroptosis-related proteins. (**G**) Effect of PTE on HC11 cell viability. Data are mean ± standard deviation (SD), ** means *p* < 0.01 compared with the control group, and ^^ means *p* < 0.01 compared with 24 h.

**Figure 3 ijms-25-10509-f003:**
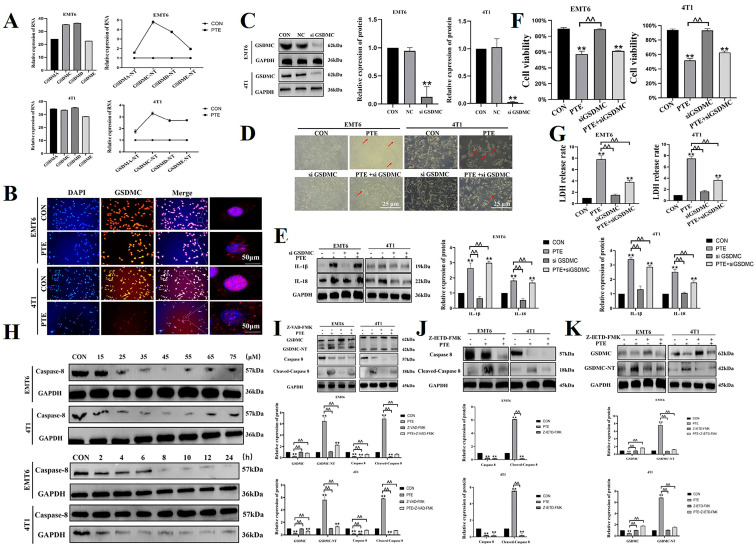
PTE induced pyroptosis through the caspase-8/GSDMC pathway. (**A**) Relative mRNA expression of GSDMA/C/D/E. (**B**) Effect of PTE on fluorescence localization of GSDMC. (**C**) Transfection efficiency of GSDMC. (**D**) The effect of GSDMC knockdown on the morphology of EMT6 and 4T1 cells under a 200-fold field of view. The red arrows pointed to cells that showed pyroptosis. (**E**) Effect of GSDMC knockdown on IL-18 and IL-1β activation in EMT6 and 4T1 cells. (**F**) Effect of GSDMC knockdown on cell viability of EMT6 and 4T1 cells. (**G**) Effect of GSDMC knockdown on LDH release of EMT6 and 4T1 cells. (**H**) Determination of Z-IETD-FMK use concentration and time. (**I**) Effect of Z-VAD-FMK on the protein expression of caspase-8 and GSDMC. (**J**) Effect of Z-IETD-FMK on caspase-8 protein expression. (**K**) Effect of Z-IETD-FMK on GSDMC protein expression. Data are mean ± SD, ** means *p* < 0.01 compared with the control group, and ^^ means *p* < 0.01 compared with the PTE group.

**Figure 4 ijms-25-10509-f004:**
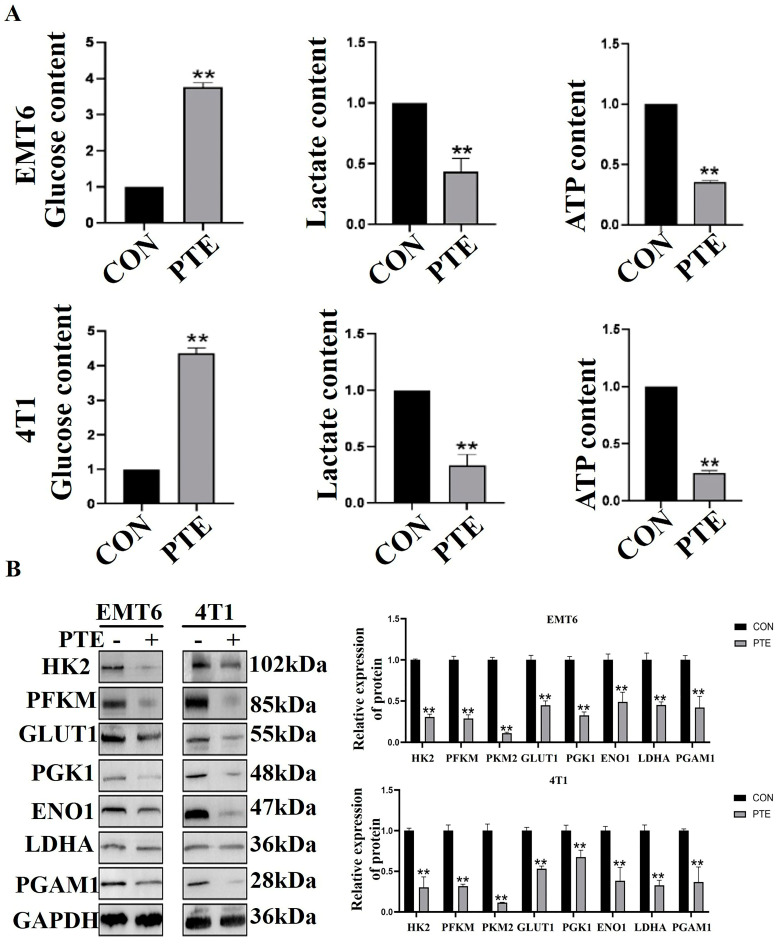
PTE inhibited EMT6 and 4T1 glycolysis. (**A**) PTE inhibited glucose consumption, lactate production, and ATP production. (**B**) PTE down-regulated the protein expression of key factors of glycolysis. Data are mean ± SD, and ** means *p* < 0.01 compared with the control group.

**Figure 5 ijms-25-10509-f005:**
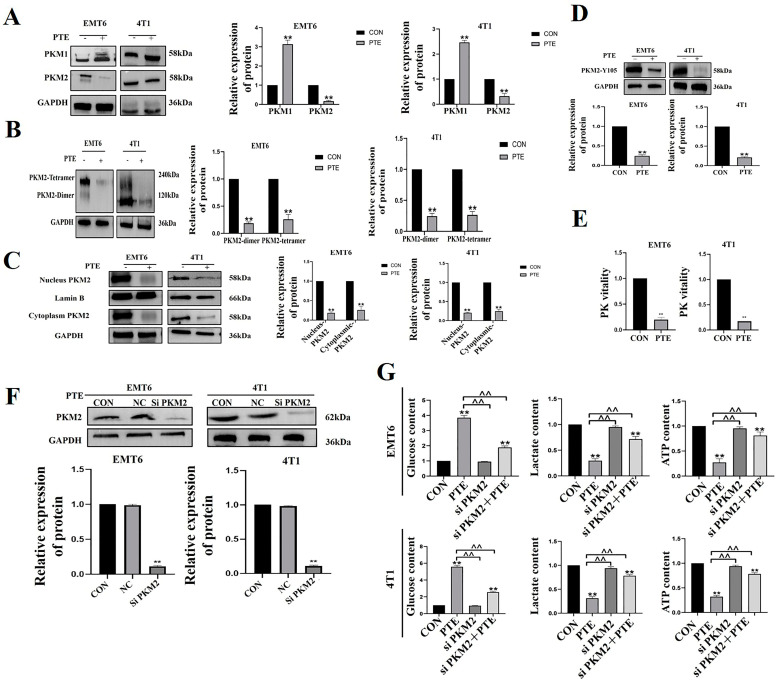
Mechanisms by which PTE regulated PKM2. (**A**) PTE regulated the expression of PKM1 and PKM2 proteins. (**B**) PTE inhibited the protein expression of PKM2 tetramer and dimer. (**C**) PTE inhibited the protein expression of nuclear PKM2. (**D**) PTE inhibited the protein expression of PKM2-Y105 phosphorylation. (**E**) Effect of PTE on pyruvate kinase activity. (**F**) Transfection efficiency of PKM2. (**G**) Glucose content, lactate content, and ATP content detection. Data are mean ± SD, ** means *p* < 0.01 compared with the control group, and ^^ means *p* < 0.01 compared with the PTE group.

**Figure 6 ijms-25-10509-f006:**
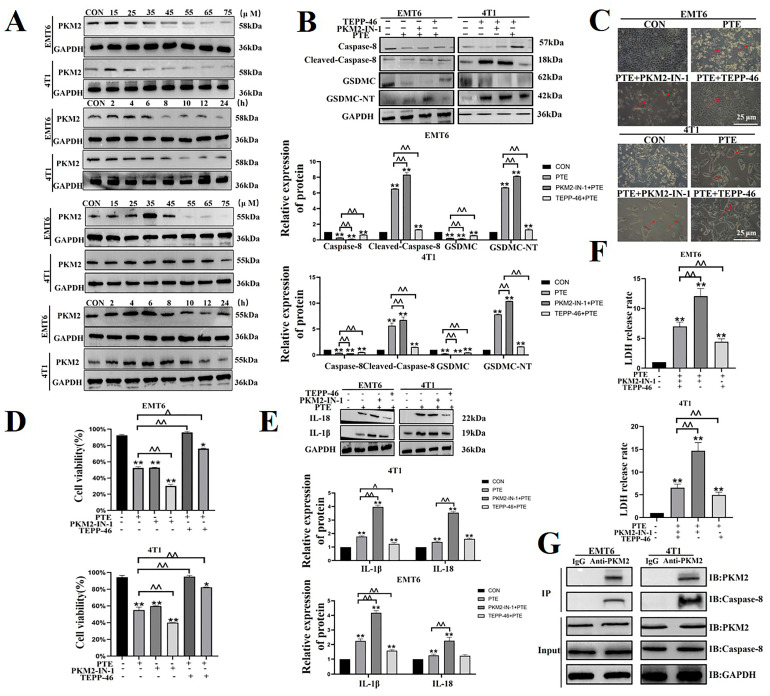
Role of PKM2 in PTE-induced pyroptosis of the caspase-8/GSDMC pathway. (**A**) Determination of TEPP-46 and PKM2-IN-1 use concentrations and timing. (**B**) Protein expression levels of caspase-8 and GSDMC. (**C**) Effect of PKM2-IN-1 or TEPP-46 combined with PTE on morphology of EMT6 and 4T1 cells under a 200-fold field of view. The red arrows pointed to cells that showed pyroptosis. (**D**) Effects of PKM2-IN-1 and TEPP-46 on cell viability. (**E**) Effects of PKM2-IN-1 and TEPP-46 on IL-18 and IL-1β activation. (**F**) Effects of PKM2-IN-1 and TEPP-46 on LDH release. (**G**) Interaction of PKM2 and caspase-8. Data are mean ± SD, ** means *p* < 0.01 compared with the control group, and ^^ means *p* < 0.01 compared with the PTE group.

**Figure 7 ijms-25-10509-f007:**
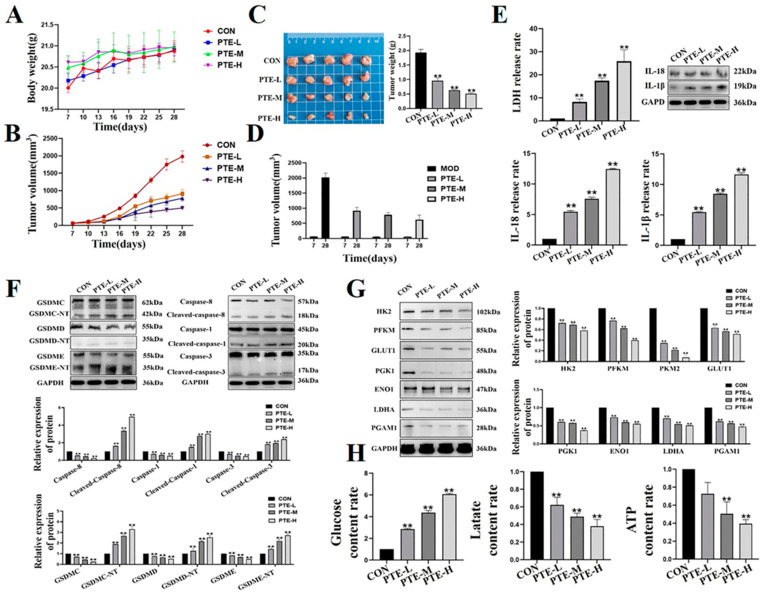
PTE inhibited tumor growth. (**A**) Effects of PTE on body weight. (**B**) Effect of PTE on tumor volume. (**C**) Effect of PTE on tumor weight. (**D**) Tumor volume on days seven and twenty-eight. (**E**) Effects of PTE on factors related to pyroptosis. (**F**) Effect of PTE on the release of LDH and activation of IL-18 and IL-1β. (**G**) Effects of PTE on glucose consumption, lactate production, and ATP production. (**H**) Effect of PTE on key factors of glycolysis. Data are mean ± SD, and ** means *p* < 0.01 compared with the control group.

**Figure 8 ijms-25-10509-f008:**
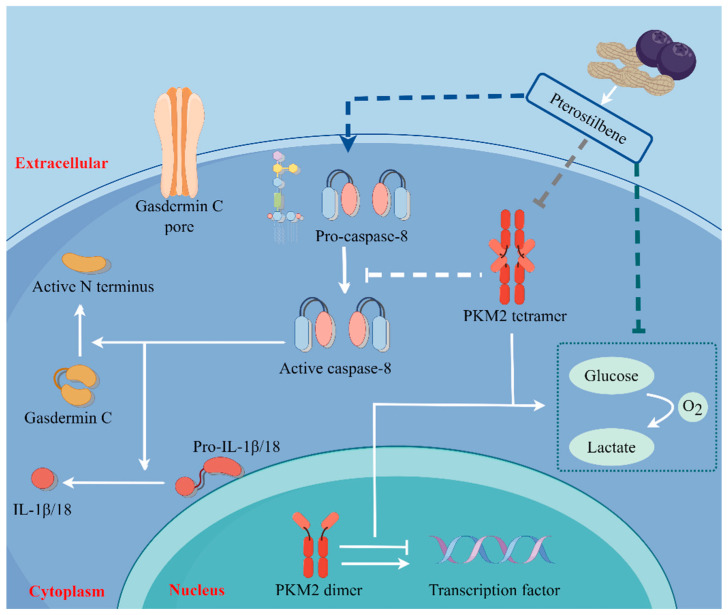
Pterostilbene induces pyroptosis in breast cancer cells through PKM2/caspase-8/GSDMC signaling pathway. Solid lines represent proven pathways, and dashed lines represent unproven pathways. The arrow represents the promoting effect, and the T-line represents the inhibiting effect.

## Data Availability

The datasets used and/or analyzed during the current study are available from the corresponding author on reasonable request.

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
