# Peer review of "Pterostilbene Induces Pyroptosis in Breast Cancer Cells through Pyruvate Kinase 2/Caspase-8/Gasdermin C Signaling Pathway"

_ijms, 2024, doi:10.3390/ijms251910509_

Round 1
Reviewer 1 Report
Comments and Suggestions for Authors
ijms-3197423
The manuscript titled "Pterostilbene Induces Pyroptosis in Breast Cancer Cells through PKM2/caspase-8/GSDMC signaling pathway" presents a detailed investigation into the mechanisms by which pterostilbene induces pyroptosis in breast cancer cells.
The study uses a good experimental approach, including cell viability assays, morphological observations, Western blot, mRNA expression analysis, and specific inhibitors to validate the findings. The research seems to be conducted in a proper way, but the editing style is poor. There are many mistakes and unnatural English style. The figures are very hard to see and evaluate.
Many data are missing. Present all the inhibitors used, for example for caspase 8, and the doses used. All effects mentioned in the text or discussion, or conclusions, should point out the dose that produces the effect.
The overall style should be more exact. For example, row 249, “PTE treatment triggered GSDMC-mediated pyroptosis in breast cancer cells”. It should mention the dose and the cells lines. It should not be generalized to all breast cells.
The authors should check all abbreviations. Each one should be declared. Even if Gasdermins (GSDMs) is presented in the abstract it should be presented again in the main text. The abbreviations should appear the first time a notion is introduced.
Row 71, PTE, C16H16O3, 256.30. It should be only Pterostilbene (PTE). No need for the molecular weight. A chemical structure should be presented. Also, on row 72, is not exactly a polyphenol. It has only one phenol group (like the authors mention). On row 78, they should also present that the additional methyl groups reduce the antioxidant capacity of PTE. See Resveratrol and Other Natural Oligomeric Stilbenoid Compounds and Their Therapeutic Applications, Plants, 2023 or similar works.
The authors should use a more scientific way when describing the methods. The style used is not proper. See for example “Add 100 microliters of ATP test solution … Leave at room temperature for 3-5 minutes…”. In this way, it sounds like a book for students.
Row 304, SPF mice? What are those?
Row 305, Regulations on the Management of Experimental Animals. Please detail what regulations? Number? Date? Organism who released the regulations?
Row 316, what was the highest concentration of DMSO used?
Row 322, OD at which wavelength?
Row 329, 7×104 pcs/mL. Check the editing
Row 349, 2h. Use an uniform style, 2h or 2 h or 2 hours, but not all of them.
Row 433, The phrase "extremely significant" is not a standard statistical term. It should be removed. It is more accurate to say that smaller P values indicate stronger evidence against the null hypothesis
The discussion seems like another abstract. It should be critical, it should point out alternative hypothesis, limitations of the study. For example, the cells are murine. Can we extend the results to human cells? Can it have other mechanisms? For example, its effect on estrogenic receptor has an influence?
Comments on the Quality of English Language
it needs corrections and better editing
Author Response
Comments 1: The study uses a good experimental approach, including cell viability assays, morphological observations, Western blot, mRNA expression analysis, and specific inhibitors to validate the findings. The research seems to be conducted in a proper way, but the editing style is poor. There are many mistakes and unnatural English style. The figures are very hard to see and evaluate.
Response 1: Thank you very much for affirming experimentation method. We have made linguistic polishes to the overall content of the article. We resized the images so that each image has a resolution greater than or equal to 300dpi.
Comments 2: Many data are missing. Present all the inhibitors used, for example for caspase 8, and the doses used. All effects mentioned in the text or discussion, or conclusions, should point out the dose that produces the effect.
Response 2: Thank you very much for your comment. We're sorry that we don't provide doses for ferroptosis inhibitor (Fer-1), necroptosis inhibitor (Nec-1), autophagy inhibitor (CQ), pyroptosis inhibitors, Caspase-8 inhibitors, PKM2 inhibitors, and PKM2 activators.By consulting the literature, Z-VAD-FMK is treated with a concentration and time of 10μM for 24 h The treatment concentration and time of Fer-1 is 5μM for 48 h The treatment concen-tration of Nec-1 is 100μM for 48 h, and the use concentration and time of CQ is 5 μM for 48 h. The Thank you very much for your comments and suggestions. We supplemented the CCK8 experiment to verify the toxicity of PTE to H711 cells of mouse mammary epithelial cells, references cited are Thank you very much for your comments and suggestions. We supplemented the CCK8 experiment to verify the toxicity of PTE to H711 cells of mouse mammary epithelial cells, references 42-45 in the article. By WB test, we determined that Z-IETD-FMK was used at a concentration and time of 45 μM for 10 h. PKM2-IN-1 was treated at a concentration and time of 65 μM for 10 h. The treatment concentration and time of TEPP-46 is 35 μM for 6 h. (row 362-368).
Comments 3: The overall style should be more exact. For example, row 249, “PTE treatment triggered GSDMC-mediated pyroptosis in breast cancer cells”. It should mention the dose and the cells lines. It should not be generalized to all breast cells.
Response 3: Thank you very much for your suggestion. For example, “PTE treatment triggered GSDMC-mediated pyroptosis in breast cancer cells”. We have changed it to " Our study is the first to suggest that treatment with PTE inhibits the expression of PKM2 to trigger pyroptosis in EMT6 and 4T1 cells via the Caspase-8/GSDMC pathway (Figure 7). Further, we showed that PTE inhibits the growth of mouse breast xenografts in vivo."(row252-255). The usage concentration of PTE is shown in the note of Figure 1. (row 94).
Comments 4: The authors should check all abbreviations. Each one should be declared. Even if Gasdermins (GSDMs) is presented in the abstract it should be presented again in the main text. The abbreviations should appear the first time a notion is introduced.
Response 4: Thank you very much for pointing out mistakes. We have checked the full text and revised them in manuscript. (row37,48,49,50,57,85,315,361,369,384,392).
Comments 5: Row 71, PTE, C16H16O3, 256.30. It should be only Pterostilbene (PTE). No need for the molecular weight. A chemical structure should be presented. Also, on row 72, is not exactly a polyphenol. It has only one phenol group (like the authors mention). On row 78, they should also present that the additional methyl groups reduce the antioxidant capacity of PTE. See Resveratrol and Other Natural Oligomeric Stilbenoid Compounds and Their Therapeutic Applications, Plants, 2023 or similar works.
Response 5: Thank you very much for pointing out mistakes. We have modified the abbreviation format of pterostilbene and inserted a picture of its chemical structure in the article. (row66-67). “Pterostilbene is a natural polyphenol” has been modified. It was replaced with this sentence “Pterostilbene (PTE) is a stilbene compound, a phenolic compound found in a variety of plants (reference 13), its chemical structure is shown in Figure A.” (row66-67). After reading Resveratrol and Other Natural Oligomeric Stilbenoid Compounds and Their Therapeutic Applications, Plants, 2023 and other relevant literature in detail, we added a description of PTE on antioxidants to the text. (row58-61).
Comments 6: The authors should use a more scientific way when describing the methods. The style used is not proper. See for example “Add 100 microliters of ATP test solution … Leave at room temperature for 3-5 minutes…”. In this way, it sounds like a book for students.
Response 6: Thank you very much for your comments and suggestions. We read a lot of literature and rewrote the Materials and methods section (row335-487). For example, you mentioned the determination of ATP content (row 451-457).
Comments 7: Row 304, SPF mice? What are those?
Response 7: Thank you very much for your comments. I'm sorry for mentioning this unprofessional term in the article, which means mice without a specific pathogen, which we abbreviate to SPF mice. But we've modified it and replaced it. (row346).
Comments 8: Row 305, Regulations on the Management of Experimental Animals. Please detail what regulations? Number? Date? Organism who released the regulations?
Response 8: Thank you very much for your comments. This experimental regulation is only the experimental management regulations of our laboratory, there is no specific number and date, from your question we have carefully found that this sentence is not suitable for the article, so we have decided to remove this sentence from the article.
Comments 9: Row 316, what was the highest concentration of DMSO used?
Response 9: Thank you very much for your comments. Based on the literature (reference 43), we confirm that DMSO is used on cells at concentrations of no more than 0.1%. (row 361-362)
Comments 10: Row 322, OD at which wavelength?
Response 10: Thank you very much for pointing out mistakes. We're sorry for the inaccuracies, we've made changes in the article (row 378).
Comments 11:Row 329, 7×104 pcs/mL. Check the editing
Response 11: Thank you very much for pointing out mistakes. We are very sorry for the inaccuracies in our statement, we have made changes in the article (row 370).
Comments 12: Row 349, 2h. Use an uniform style, 2h or 2 h or 2 hours, but not all of them.
Response 12: Thank you very much for pointing out mistakes. We have revised the full text and have all been unified to 2 h (Such as row 363-368).
Comments 13: Row 433, The phrase "extremely significant" is not a standard statistical term. It should be removed. It is more accurate to say that smaller P values indicate stronger evidence against the null hypothesis.
Response 13: Thank you very much for pointing out mistakes. We have revised the article in full as you suggested, adding P values after each result. And the phrase "extremely significant" was removed.
Comments 14: The discussion seems like another abstract. It should be critical, it should point out alternative hypothesis, limitations of the study. For example, the cells are murine. Can we extend the results to human cells? Can it have other mechanisms? For example, its effect on estrogenic receptor has an influence?
Response 14: Thank you very much for your comments. We have substantially revised the discussion. We discussed the different hypotheses you mentioned and the limitations of the study (row 295-297, 314-317, 325-327). PTE has been shown to have an effect on human breast cancer, for example, PTE can inhibit triple-negative breast cancer (reference 19), so we believe that our results can be extended to human breast cancer cells, which needs to be confirmed by subsequent studies. It has been reported that PTE can induce cancer cell death through other mechanisms. For example, PTE can induce apoptosis of cancer cells through the following pathways: p53/cyclinE1, Bcl-2/caspase-3/caspase-9 and caspase-8 signaling pathways (row 281-283, Thank you very much for your comments and suggestions. We supplemented the CCK8 experiment to verify the toxicity of PTE to H711 cells of mouse mammary epithelial cells, references 23-25). The estrogen receptor you mention is a key tumor signaling molecule through which PTE can suppress cancer (row 279-282, Thank you very much for your comments and suggestions. We supplemented the CCK8 experiment to verify the toxicity of PTE to H711 cells of mouse mammary epithelial cells, references 28-30).
Reviewer 2 Report
Comments and Suggestions for Authors
The manuscript ‘’Pterostilbene Induces Pyroptosis in Breast Cancer Cells through PKM2/caspase-8/GSDMC signaling pathway’’ examines the anti-cancer activity of pterostilbene (PTE), a natural polyphenolic compound, and its capacity to induce pyroptotic cell death in breast cancer cells via suppression of glycolysis. Pyroptosis is an inflammatory form of programmed cell death, and the findings published in this manuscript demonstrate that PTE activates pyroptosis via the PKM2/caspase-8/GSDMC pathway. In the present study, PTE demonstrates a reduction of cancer cell viability and a reduction in tumor growth in experimental models with the discovery of a statistically significant decrease in the expression of genes involved in glycolysis and a significant increase in genes that induce pyroptosis.
The study is significant as it sheds new light on the ability for PTE to induce pyroptosis in breast cancer cells, specifically via the PKM2/caspase-8/GSDMC pathway and glycolysis inhibition. This research may ultimately lead to the design of new therapeutics for breast cancer that may attenuate patients' complaint of refractory cancers after chemotherapy.
I believe this manuscript is worthy of publication if important points are addressed:
1. Broader evaluation of cell death pathways: The study is focused mainly on the PKM2/caspase-8/GSDMC pathway for triggering pyroptosis in breast cancer cells. It would be beneficial to extend the scope to other cell death pathways, including apoptosis and necroptosis. A more extensive evaluation of the diverse pathways will provide a panoramic view of the beneficial effects of PTE. That is to say, understanding how PTE interacts with various forms of cell death may assist with the understanding of the complex mechanisms by which PTE operates.
2. Generalizability of results to other types of cancer: The findings from this study could also be generalized to other types of cancer besides breast cancer. It would be informative to examine the action of PTE in other cancer lines to determine if the results are universally applicable or specific to breast cancer. This would increase the generalizability of the results and could lead to more broad-based therapeutic strategies using PTE.
3. More detailed analysis of PTE toxicity: Despite noting in the discussion that PTE was not toxic at two weeks in experimental animals in terms of mortality, further work studying toxicity in normal cells and tissues should be performed. The studies noted toxicity of PTE in normal body cells and long-type animal studies are recommended. This would demonstrate PTE's comfort in being safe for clinical applications, conversely allowing to mitigate the chance of unwanted side effects.
4. Comparative analysis with other anticancer drugs: Further, including a comparative analysis will increase the impact of efficacy of PTE to those with similar pathways or pyropotic fashions. That way, PTE will be situated in the context of current therapeutic practice and establish PTE's superiority or related advantages.
5. Further explanation of the methodology: The methods used in the research could be elaborated, adding more detail about culture conditions of cells, determine how data are analyzed, analysis methods used, controls used, etc. This would assist the reader to better visualize experimental methods and promote reproducibility.
6. English language: In general, the manuscript is articulately expressed and understandable. Sentences are coherent and relevant technical terms are used appropriately. Nonetheless, several opportunities exist to provide minor modifications to enhance the fluency of the text to make it more explicit.
Comments on the Quality of English Language
Minor editing of English language required.
Author Response
Comments 1: Broader evaluation of cell death pathways: The study is focused mainly on the PKM2/caspase-8/GSDMC pathway for triggering pyroptosis in breast cancer cells. It would be beneficial to extend the scope to other cell death pathways, including apoptosis and necroptosis. A more extensive evaluation of the diverse pathways will provide a panoramic view of the beneficial effects of PTE. That is to say, understanding how PTE interacts with various forms of cell death may assist with the understanding of the complex mechanisms by which PTE operates.
Response 1: Thank you very much for your comments and suggestions. First of all, before doing the research, we found that PTE can kill cells in other cancers through other death methods by reviewing a lot of literature, which we added to the discussion section (row 272-274,282-283). Based on this literature, we used a variety of death inhibitors in combination with PTE to determine the specific mode of PTE-induced death of EMT6 and 4T1 cells. The experimental data showed that the mode of death induced by PTE was pyroptosis. Existing studies and our data show that PTE is a clinical potential agent that can regulate multiple deaths through different mechanisms.
Comments 2: Generalizability of results to other types of cancer: The findings from this study could also be generalized to other types of cancer besides breast cancer. It would be informative to examine the action of PTE in other cancer lines to determine if the results are universally applicable or specific to breast cancer. This would increase the generalizability of the results and could lead to more broad-based therapeutic strategies using.
Response 2: Thank you very much for your comments and suggestions. Many studies have confirmed the role of PTE in other cancers, which we have elaborated in addition to the discussion (row 268-270). We were inspired to wonder if PTE has a role in breast cancer in mice. The final data showed that PTE induced pyroptosis in EMT6 and 4T1 cells. In the future, we will strive to explore whether PTE can affect the progression of other types of tumors through the pyrogenic pathway, and strive to conduct practical studies as soon as possible(row 272-274,282-283).
Comments 3: More detailed analysis of PTE toxicity: Despite noting in the discussion that PTE was not toxic at two weeks in experimental animals in terms of mortality, further work studying toxicity in normal cells and tissues should be performed. The studies noted toxicity of PTE in normal body cells and long-type animal studies are recommended. This would demonstrate PTE's comfort in being safe for clinical applications, conversely allowing to mitigate the chance of unwanted side effects.
Response 3: Thank you very much for your comments and suggestions. We supplemented the CCK8 experiment to verify the toxicity of PTE to H711 cells of mouse mammary epithelial cells, and the results showed that PTE had no proliferative toxicity to H711 cells (Figure 1H). In addition, the protective effects of PTE on cardiomyocytes, liver cells, and kidney cells were added to the discussion (row 261-267, Thank you very much for your comments and suggestions. We supplemented the CCK8 experiment to verify the toxicity of PTE to H711 cells of mouse mammary epithelial cells, references 24-28). Previous studies and our results indicate that PTE is safe at the cellular level. In addition, studies have confirmed the safety of PTE in rats (row 258-261, reference 24), and our data also confirm the high safety of PTE in mice. Based on the results of previous studies and this study, we believe that PTE has the potential to be a safe and effective clinical therapy. Of course, more detailed toxicity analysis is also valuable, such as the toxicity analysis after long-term action of drugs in mice, which we will strive to complete in the follow-up study, but the existing results have proved that PTE has a certain safety.
Comments 4: Comparative analysis with other anticancer drugs: Further, including a comparative analysis will increase the impact of efficacy of PTE to those with similar pathways or pyroptosis fashions. That way, PTE will be situated in the context of current therapeutic practice and establish PTE's superiority or related advantages.
Response 4: Thank you very much for your comments and suggestions. At present, the commonly used chemotherapy drugs have high toxic side effects (such as anthracyclines), so it is very important to study the effects of plant chemotherapy drugs with high safety. Phenolic drugs are the most studied phytochemotherapeutic drugs at present, but the other phenolic drugs are also in the basic research stage, so the comparison is of little significance. However, we have benefited a lot from your comments, and we will add corresponding positive controls in subsequent studies. In this article, we have included the disadvantages of other commonly used phenolic compounds compared with PTE in the discussion section. (row 256-261, reference 22-23).
Comments 5: Further explanation of the methodology: The methods used in the research could be elaborated, adding more detail about culture conditions of cells, determine how data are analyzed, analysis methods used, controls used, etc. This would assist the reader to better visualize experimental methods and promote reproducibility.
Response 5: Thank you very much for your comments and suggestions. We are very sorry for the lack of details in this part. We have consulted a lot of literature to learn their writing format in this part, readjust the writing in this part and added the details of cell culture conditions, drug dosage, test group and analysis method. (row 335-479).
Comments 6: English language: In general, the manuscript is articulately expressed and understandable. Sentences are coherent and relevant technical terms are used appropriately. Nonetheless, several opportunities exist to provide minor modifications to enhance the fluency of the text to make it more explicit.
Response 6: Thank you very much for your comments and suggestions. We have made some changes to the less appropriate language in the article.
Round 2
Reviewer 2 Report
Comments and Suggestions for Authors
The authors made all suggested corrections to the manuscript. The revised version agrees with me. I recommend publishing the article.
Academic Editor
Comments and Suggestions for Authors
Two reviewers pointed out important issues, and it is judged that the authors have made appropriate corrections accordingly. However, there are a few additional issues that need to be further revised.
[Major concerns]
1. Cell line: The authors have provided detailed descriptions only for EMT6 and 4T1 cells, but the culture methods for mammary epithelial cells are missing. Additionally, in Figure 1H, the cell name is labeled as HC11, while in the main text (lines 88 and 89), it is listed as H711. Please correct this discrepancy and provide detailed information on the culture methods for these cells.
2. English: The content of the research is excellent, but more careful attention needs to be paid to writing the English manuscript. The sentences revised after the reviewers' comments are not smooth. Additionally, in the case of common chemical compounds, which are not proper nouns, their names should not be capitalized in the middle of a sentence. However, in too many instances, they are unnecessarily capitalized. Therefore, it is essential to have the manuscript proofread by a professional English editor before final acceptance for publication.
3. Notations of certain unit: It is a rule that there should be a space between a number and its unit, but this rule has not been followed in several places. Please find and correct all such instances.
4. Pterostilbene: It would be helpful to include the chemical structure of pterostilbene, trans-3,5-dimethoxy-4-hydroxystilbene, in the introduction section of the paper. Additionally, it is important to mention in the discussion that the double bonds in Pterostilbene are photosensitive, making it highly susceptible to photodegradation, which raises stability concerns. Research is also underway to develop new drugs that address this photodegradation issue
5. Figure A: I believe Figure A may have been inserted during the review process. However, it is standard practice in IJMS to number figures sequentially using Arabic numerals. Please correct this accordingly.
6. Abbreviations: The use of abbreviations when writing a paper has many advantages besides simplicity of expression. To use an abbreviation, first write the abbreviation in parentheses after the full name, and then use the abbreviation from Introduction to the final Conclusion. Abbreviations should only be used if they are repeatedly used and if they are not used again, only the full name should be used. In particular, because of the characteristics of IJMS, where Materials and Methods is arranged at the end of the paper, the original words and abbreviations are written in the order they are used from the introduction, and only when the abbreviation is used repeatedly, the abbreviation can be used until the conclusion.
7. In cases where abbreviations are used within figures or tables, please list these abbreviations along with their corresponding full names in the figure legends or at the bottom of corresponding tables. If there are two or more abbreviations, arrange them in alphabetical order.
8. IL-1β and IL-18: When referring to these two interleukins in the text, they are written as either "IL-1β and IL-18" or "IL-18 and IL-1β." Unless there is a specific reason, it would be better to consistently use the pattern "IL-1β and IL-18" throughout the paper.
9. Figure 2H: When labeling the various concentrations or time points above the WB in Figure 2H, the lack of space due to listing the units of concentration and time together with the numbers makes it difficult to fit within the width of each WB lane. I suggest listing the numbers, and then placing "(μM)" or "(h)" at the far-right end. This should provide more space for clear labeling. Applying the same method to the other figures will provide more space and improve the overall appearance.
10. Figure 6C: In Figure 6C, the left image shows the order PTE-L, PTE-H, PTE-M, while the right graph shows the order PTE-L, PTE-M, PTE-H. I understand this discrepancy might have occurred due to an error when taking the left image, but at the very least, the right graph in Figure 6C should follow the same order as the left image.
[Minor concerns]
1. Line 333: pKM2 should be written as PKM2.
2. Line 347: Arabic numerals should not appear as the first word in an English sentence. Please make the necessary corrections accordingly.
3. Line 348: Re-write ‘Biotech-nology’.
4. Line 349: Re-write the temperature unit.
5. Line 386: Re-write this sentence.
6. Line 402: Re-write the company name, thermo.
7. Line 468: Re-write Na3VO4.
8. Line 489: If abbreviations are unnecessary, but must be listed, please arrange them in alphabetical order and capitalize the first letter of each.
9. References: Please revise the references to conform to the format required by IJMS, as the following references are missing page number. Examples: 13, 22, 23, 25, 29, 31, 36, etc.
Overall, the manuscript can be considered to publication after minor revision as indicated above.
Author Response
Comments 1: Cell line: The authors have provided detailed descriptions only for EMT6 and 4T1 cells, but the culture methods for mammary epithelial cells are missing. Additionally, in Figure 1H, the cell name is labeled as HC11, while in the main text (lines 88 and 89), it is listed as H711. Please correct this discrepancy and provide detailed information on the culture methods for these cells.
Response 1: Thank you for pointing out our mistake. The correct name of mouse mammary epithelial cell is HC11, and we have modified "H711" in the article (line 91 and 92). We have added HC11 culture method to the materials and methods (lines 331,332).
Comments 2: English: The content of the research is excellent, but more careful attention needs to be paid to writing the English manuscript. The sentences revised after the reviewers' comments are not smooth. Additionally, in the case of common chemical compounds, which are not proper nouns, their names should not be capitalized in the middle of a sentence. However, in too many instances, they are unnecessarily capitalized. Therefore, it is essential to have the manuscript proofread by a professional English editor before final acceptance for publication.
Response 2: Thank you for pointing out our mistake. We retouched the rough sentences in the article (The red font in the article). For common compounds that are not proper nouns in the article, we have changed them to lower case (lines 53,54,346,381).
Comments 3: Notations of certain unit: It is a rule that there should be a space between a number and its unit, but this rule has not been followed in several places. Please find and correct all such instances.
Response 3: Thank you for pointing out our error, we have checked and corrected the whole text (lines 97, 342, 343, 359, 360, 361, 362, 363, 367, 368, 369, 370, 371, 372, 373, 377, 382, 399, 409, 410, 412, 413, 418, 419, 420, 436, 438, 439, 440, 444, Font on yellow background in article).
Comments 4: Pterostilbene: It would be helpful to include the chemical structure of pterostilbene, trans-3,5-dimethoxy-4-hydroxystilbene, in the introduction section of the paper. Additionally, it is important to mention in the discussion that the double bonds in Pterostilbene are photosensitive, making it highly susceptible to photodegradation, which raises stability concerns. Research is also underway to develop new drugs that address this photodegradation issue
Response 4: We have added a discussion of photosensitivity to the discussion (lines 261-264).
Comments 5: Figure A: I believe Figure A may have been inserted during the review process. However, it is standard practice in IJMS to number figures sequentially using Arabic numerals. Please correct this accordingly.
Response 5: Thank you for pointing out our mistake. We have changed Figure A to Figure 1(line 71).
Comments 6: Abbreviations: The use of abbreviations when writing a paper has many advantages besides simplicity of expression. To use an abbreviation, first write the abbreviation in parentheses after the full name, and then use the abbreviation from Introduction to the final Conclusion. Abbreviations should only be used if they are repeatedly used and if they are not used again, only the full name should be used. In particular, because of the characteristics of IJMS, where Materials and Methods is arranged at the end of the paper, the original words and abbreviations are written in the order they are used from the introduction, and only when the abbreviation is used repeatedly, the abbreviation can be used until the conclusion.
Response 6: Thank you for your suggestions, we have checked and modified the whole text.
Comments 7: In cases where abbreviations are used within figures or tables, please list these abbreviations along with their corresponding full names in the figure legends or at the bottom of corresponding tables. If there are two or more abbreviations, arrange them in alphabetical order.
Response 7: Thank you for your suggestion. We have rearranged the table of abbreviations alphabetically (lines 92-93).
Comments 8: IL-1β and IL-18: When referring to these two interleukins in the text, they are written as either "IL-1β and IL-18" or "IL-18 and IL-1β." Unless there is a specific reason, it would be better to consistently use the pattern "IL-1β and IL-18" throughout the paper.
Response 8: Thank you very much for your suggestion. We unify them as "IL-18 and IL-1β"(lines 123,204,224,225)
Comments 9: Figure 2H: When labeling the various concentrations or time points above the WB in Figure 2H, the lack of space due to listing the units of concentration and time together with the numbers makes it difficult to fit within the width of each WB lane. I suggest listing the numbers, and then placing "(μM)" or "(h)" at the far-right end. This should provide more space for clear labeling. Applying the same method to the other figures will provide more space and improve the overall appearance.
Response 9: Thank you for your advice. We have modified it (line 140).
Comments 10: In Figure 6C, the left image shows the order PTE-L, PTE-H, PTE-M, while the right graph shows the order PTE-L, PTE-M, PTE-H. I understand this discrepancy might have occurred due to an error when taking the left image, but at the very least, the right graph in Figure 6C should follow the same order as the left image.
Response 10: Thank you very much for pointing out our mistake. We have modified it (line 231).
[Minor concerns]
- Line 333: pKM2 should be written as PKM2 (line 323).
- Line 347: Arabic numerals should not appear as the first word in an English sentence. Please make the necessary corrections accordingly (line 340).
- Line 348: Re-write ‘Biotech-nology’ (line 351).
- Line 349: Re-write the temperature unit (line 372).
- Line 386: Re-write this sentence (lines 381,382).
- Line 402: Re-write the company name, thermos (line 397).
- Line 468: Re-write Na3VO4 (line 451).
- Line 489: If abbreviations are unnecessary, but must be listed, please arrange them in alphabetical order and capitalize the first letter of each (lines 492,493).
- References: Please revise the references to conform to the format required by IJMS, as the following references are missing page number. Examples: 13, 22, 23, 25, 29, 31, 36, etc. (References 13,22,24,25,26,29,30,31,37).
Response: Thank you for pointing out our mistake. All errors have been fully corrected (lines 232,340,351,372,381,382,397,451,492,493. References 13,22,24,25,26,29,30,31,37).